# Functional and Transcriptional Adaptations of Blood Monocytes Recruited to the Cystic Fibrosis Airway Microenvironment In Vitro

**DOI:** 10.3390/ijms22052530

**Published:** 2021-03-03

**Authors:** Bijean D. Ford, Diego Moncada Giraldo, Camilla Margaroli, Vincent D. Giacalone, Milton R. Brown, Limin Peng, Rabindra Tirouvanziam

**Affiliations:** 1Department of Pediatrics, Emory University School of Medicine, Atlanta, GA 30322, USA; bdford@emory.edu (B.D.F.); dmoncad@emory.edu (D.M.G.); vincent.giacalone@emory.edu (V.D.G.); mbrow05@emory.edu (M.R.B.); 2Center for CF & Airways Disease Research, Children’s Healthcare of Atlanta, Atlanta, GA 30322, USA; 3Department of Medicine, University of Alabama at Birmingham, Birmingham, AL 35233, USA; cmargaroli@uabmc.edu; 4Department of Biostatistics and Bioinformatics, Emory University, Atlanta, GA 30322, USA; lpeng@emory.edu

**Keywords:** bacteria, lung disease, scavenging, transmigration

## Abstract

Cystic fibrosis (CF) lung disease is dominated by the recruitment of myeloid cells (neutrophils and monocytes) from the blood which fail to clear the lung of colonizing microbes. In prior in vitro studies, we showed that blood neutrophils migrated through the well-differentiated lung epithelium into the CF airway fluid supernatant (ASN) mimic the dysfunction of CF airway neutrophils in vivo, including decreased bactericidal activity despite an increased metabolism. Here, we hypothesized that, in a similar manner to neutrophils, blood monocytes undergo significant adaptations upon recruitment to CFASN. To test this hypothesis, primary human blood monocytes were transmigrated in our in vitro model into the ASN from healthy control (HC) or CF subjects to mimic in vivo recruitment to normal or CF airways, respectively. Surface phenotype, metabolic and bacterial killing activities, and transcriptomic profile by RNA sequencing were quantified post-transmigration. Unlike neutrophils, monocytes were not metabolically activated, nor did they show broad differences in activation and scavenger receptor expression upon recruitment to the CFASN compared to HCASN. However, monocytes recruited to CFASN showed decreased bactericidal activity. RNASeq analysis showed strong effects of transmigration on monocyte RNA profile, with differences between CFASN and HCASN conditions, notably in immune signaling, including lower expression in the former of the antimicrobial factor ISG15, defensin-like chemokine CXCL11, and nitric oxide-producing enzyme NOS3. While monocytes undergo qualitatively different adaptations from those seen in neutrophils upon recruitment to the CF airway microenvironment, their bactericidal activity is also dysregulated, which could explain why they also fail to protect CF airways from infection.

## 1. Introduction

Cystic fibrosis (CF) is a fatal autosomal recessive disease affecting an estimated 80,000 individuals worldwide [1]. CF is caused by mutations of the CF transmembrane conductance regulator (CFTR) anion channel, of which more than 2000 are recognized [2]. CFTR dysfunction perturbs the function of exocrine epithelia across the body, with lung disease being the primary cause of morbidity and mortality among patients [3]. CF lung disease is associated with an early and massive recruitment of neutrophils from the blood, which, however, fails to control colonizing microorganisms, such as the bacteria *Staphylococcus aureus* (*S. aureus*) and *Pseudomonas aeruginosa* (*P. aeruginosa*) [4]. This paradox was explained by our discovery of a novel fate of human neutrophils developed after transmigration into the CF airway microenvironment. This fate is observable in vivo [5,6,7] and in an in vitro transmigration model that we developed [8] and causes these professional phagocytes to increase metabolic activity and granule exocytosis, while decreasing their bactericidal activity.

In addition to neutrophils, monocytes from the blood are also recruited to CF airways [9]. In other chronic diseases, such as chronic obstructive pulmonary disease (COPD) [10] and rheumatoid arthritis [11], blood monocytes recruited to peripheral tissues (lung and joint, respectively) have been shown to adapt to local microenvironments and alter their functional and transcriptional profiles [12]. Thus, we investigated here potential alterations to monocyte function upon recruitment to the CF airway microenvironment. To this end, we used our lung transmigration model, validated previously with neutrophils [8,13,14], in which primary blood leukocytes are transmigrated through a well-differentiated epithelium into the apical airway fluid supernatant (ASN) from patients with CF or healthy controls (HC). Primary blood monocytes were thus made to transmigrate in our model, after which their surface phenotype, metabolic and bacterial killing activities, and transcriptomic profile by RNA sequencing (RNASeq) were quantified.

## 2. Results

### 2.1. Blood Monocyte Transmigration into CFASN vs. HCASN Results in Similar Yield and Viability But Different Bactericidal Abilities

Blood monocytes from CF and HC subjects were transmigrated across a well-differentiated human airway epithelium at the air–liquid interface (ALI), into either CFASN or HCASN, and analyzed after 10 h for functional and transcriptional properties (Figure 1A). Gating and absolute counting of live transmigrated monocytes was achieved by flow cytometry (Appendix A). We did not observe any significant difference in the cell yield or viability of the transmigrated monocyte pools based on the origin of the blood monocytes or ASN used (Figure 1B). However, migration into CFASN vs. HCASN significantly decreased the ability of both CF and HC monocytes to kill the Gram-positive organism *S. aureus* (Figure 1C). Migration into CFASN vs. HCASN also significantly decreased the ability of CF monocytes to kill the Gram-negative organism *P. aeruginosa.* However, HC monocytes migrated into the CFASN displayed increased killing of *P. aeruginosa* compared to those migrated to HCASN.

### 2.2. Blood Monocyte Transmigration into CFASN vs. HCASN Results in Similar Metabolic Activity

Next, we collected CF and HC blood monocytes transmigrated to CFASN or HCASN and subjected them to analysis of mitochondrial respiration using a Seahorse XFp system. We derived from the oxygen consumption rate (OCR) measure (Figure 2A) five distinct metrics: basal respiration (Figure 2B), ATP-linked respiration (Figure 2C), maximal respiratory capacity (Figure 2D), reserve capacity (Figure 2E), and non-mitochondrial respiration (Figure 2F). We did not observe any difference in the five metrics based on the origin of blood monocytes or ASN used.

### 2.3. Blood Monocyte Transmigration into CFASN vs. HCASN Results in Minimal Differences in Expression of Scavenger Receptors and Activation Markers

We next turned our attention to surface phenotyping. After transmigration, scavenger receptors CD36, CD91, CD47 and its cognate receptor CD172a did not show a unifying pattern in their change in expression based on the origin of the monocytes and ASN (Figure 3 for aggregated statistics and Appendix A for representative histograms).

CD36 and CD91 were only significantly increased compared to baseline in CF monocytes transmigrated to HCASN. CD47 and CD172a were significantly increased compared to baseline in CF monocytes transmigrated to CFASN and HCASN, with significant differences between HC monocytes transmigrated to CFASN vs. HCASN for both.

After transmigration, activation markers CD63, HLA-DR (class II major histocompatibility complex), programmed death-1 (PD-1) and its cognate receptor programmed death-ligand 1 (PD-L1) did not show a unifying pattern in their change in expression based on the origin of monocytes and ASN (Figure 4 for aggregated statistics and Appendix A for representative histograms).

CD63 was increased compared to baseline for both CF and HC monocytes transmigrated to CFASN and HCASN, but there was no difference between CF and HC monocytes within each ASN condition, or between ASN conditions within each type of monocyte; thus, the process of transmigration was responsible for significantly increasing CD63 expression. HLA-DR was only significantly increased compared to baseline to HC monocytes transmigrated to CFASN. PD-1 was increased compared to baseline in CF monocytes transmigrated to HCASN and HC monocytes transmigrated to CFASN, with a significant difference between CF and HC monocytes transmigrated to HCASN. Finally, PD-L1 was increased compared to baseline in CF monocytes transmigrated to CFASN and HC monocytes transmigrated to both CFASN or HCASN, and HC monocytes transmigrated to HCASN.

### 2.4. Blood Monocyte Transmigration into CFASN vs. HCASN Results in Significant Differences in Transcriptional Profile

In contrast with metabolism and surface markers, there were significant transcriptional differences between CFASN and HCASN transmigration conditions. According to principal component analysis (Figure 5A), there was complete discrimination in the transcriptional profiles of HC monocytes comparing baseline (pre-transmigration), HCASN, and CFASN transmigration conditions. As visualized in volcano plots (Appendix A) and pathway analyses for differentially expressed genes (DEGs) in monocytes transmigrated to HCASN (Figure 5B) and CFASN (Figure 5C), respectively (both in comparison to baseline), multiple critical pathways were significantly different between the two conditions. These included pathways related to cell signaling, contractility, cytoskeleton, immune response, mucin expression, and stem cell regulation.

To better visualize which pathways were commonly or uniquely altered in CFASN and HCASN transmigration conditions, Venn diagrams were computed based on significantly dysregulated genes. The resulting gene signatures were overlapped to KEGG Biocarta, Reactome, and Hallmark gene sets. As a result, we identified 3578 shared upregulated genes (Appendix A) associated with small-molecular transport, post-translational modification, and lipid metabolism, whose upregulation was caused by the process of transmigration per se. Monocytes transmigrated to HCASN uniquely upregulated 1863 genes related to transcription and translation (Appendix A), while those transmigrated to CFASN uniquely upregulated 740 genes related to intracellular trafficking and transport and cytokine signaling (Appendix A). We also identified 3254 shared downregulated genes (Appendix A) related to translation and lipid metabolism. This matches the scale of change and pathways seen for shared upregulated genes, indicative of the bidirectional regulation of these functions upon transmigration. Monocytes transmigrated to HCASN uniquely downregulated 1244 genes related to oxidative metabolism and innate immunity (Appendix A), while those transmigrated to CFASN uniquely downregulated 1616 genes related to translation (Appendix A). 

To investigate whether the altered antibacterial activity of monocytes transmigrated to CFASN may be associated with changes in subcategories of immune response genes, a directed analysis was conducted (Figure 6).

Genes involved in antigen presentation (Figure 6A) and encoding interleukins (Figure 6B) and cytokines/chemokines (Figure 6C) showed very similar profiles in HCASN and CFASN transmigrated monocytes, with the exception of CXCL11 (increased in the HCASN condition and decreased in the CFASN condition) and CCR1 (decreased in the HCASN condition and increased in the CFASN condition). There were more pronounced differences in the expression pattern of antimicrobial genes (Figure 6D), notably interferon-stimulated gene 15 (ISG15) and C-X-C motif chemokine 11 (CXCL11), two critical effectors of antibacterial immunity which were significantly downregulated in monocytes transmigrated to CFASN (Figure 6E). Comparing genes encoding nitric oxide (NO) synthase (NOS) enzymes, we found significantly higher induction of NOS3 in HCASN vs. CFASN transmigrated monocytes (Figure 6F). This increased expression of NOS3 coincided with the increased production of intracellular NO in the former (Figure 6F).

For completeness and to enable independent reanalysis of our RNASeq data, we posted transcriptomic comparisons between blood monocytes, HCASN-recruited monocytes, and CFASN-recruited monocytes in an online repository at: https://data.mendeley.com/datasets/gcp66ch34c/3 (accessed on 23 February 2021)

## 3. Discussion

In this study, we mimicked the recruitment of primary human blood monocytes to CF vs. HC airways using a recently developed model [8,13,14] which uses a well-differentiated small airway epithelium and human airway supernatant as conditioning medium for transmigrated leukocytes. Our findings suggest that monocytes transmigrated into CFASN do not demonstrate broadly different changes in their surface phenotype and metabolism. However, they are less efficient at killing bacteria. Transcriptional profiling suggests broader changes in monocytes migrated to CFASN, notably in antimicrobial poise.

While previous studies have explored the role of monocytes in CF through analysis of those cells in blood [15,16,17,18], understanding their role in airway inflammation has been challenging, due to their low yield in airway samples and the practical difficulties of replicating this microenvironment in a physiologically relevant manner. To our knowledge, our transmigration model is unique in its ability to enable the migration and conditioning of naïve primary human blood monocytes in primary airway fluid supernatant from patients with CF and control subjects. We initially focused on the analysis of the surface phenotype, metabolism, and bacterial killing of transmigrated monocytes, since all these functional correlates were altered in the context of CFASN for transmigrated neutrophils [8].

For surface phenotyping, we focused our attention on select activation markers important in the extrusion of monocyte granules (CD63) [19], antigen presentation (HLA-DR), and immune poise (PD-1 and PD-L1 [20,21]), and select scavenger receptors involved in the uptake of apoptotic cells, small vesicles, and debris, such as CD36 [22,23], CD91 [24,25], CD47 [26], and its binding partner CD172a [27]. While transmigration to either CFASN or HCASN altered the expression of several of these markers compared to baseline (blood monocytes pre-transmigration), only CD47 and CD172 were different between the two transmigration conditions, and only when HC monocytes were used in the transmigration. A broad lack of difference in metabolic activity based on real-time analysis of mitochondrial respiration was also observed between the two conditions. Therefore, monocytes do not undergo the same functional activation observed previously for neutrophils recruited in the CF airway environment in vitro [8] and in vivo [5,6,7]. This is consistent with the seemingly limited role of these cells in CF airway pathogenesis [3].

Nevertheless, because of their built-in bactericidal abilities [28], it remains possible that recruited monocytes may play a role, even minimal, in the control of bacterial pathogens present in CF airways. Indeed, we did observe a significant decrease in the bacterial killing of monocytes transmigrated to CFASN compared to HCASN. This was true for both HC and CF monocytes in the case of *S. aureus* (Gram-positive organism) and CF monocytes in the case of *P. aeruginosa* (Gram-negative organism). Since this discrepancy was not explained by the altered expression of the select activation markers and scavenger receptors that we had surveyed or by the differential metabolic activity, we next turned our attention to potential differences in transcriptional profiles, to gain insight into other potential pathways of interest.

We observed that most pathways found to be different in multiple immune-related genes and pathways were primarily regulated by transmigration (i.e., they were shared between CFASN and HCASN conditions compared to the blood baseline pre-transmigration). Approximately 7800 genes were shared between the two transmigration conditions, around half being upregulated (3578) and half downregulated (3254), as compared to blood. However, there were also significant differences between monocytes transmigrated to CFASN and HCASN, notably in transcriptional and translational regulation, lipid metabolism, and immune response genes. Among these, several interferon-response genes, such as the multifunctional factor ISG15 [29] and chemokine CXCL11 [30], were specifically upregulated in monocytes transmigrated to HCASN and downregulated in those recruited to CFASN. ISG15 is a ubiquitin-like molecule with broad roles as an intracellular regulator of host and microbial proteins and an extracellular ligand to the CD11a/CD18 coreceptor [31], with strong positive effects on immune cell chemoattraction and activation, and therefore on antibacterial responses, notably against *P. aeruginosa* [32]. CXCL11 is a chemokine able to attract immune cells to a peripheral site, but also doubles as a direct antimicrobial factor, with defensin-like properties [33]. As another example, we also found evidence for the lower expression of NO synthase 3 (which was shown to be critical to monocyte function in certain conditions [34,35,36]) and lower intracellular NO levels in monocytes transmigrated to CFASN compared to HCASN. Prior studies have shown that NO levels are decreased in CF airways [37], in part because of increased extracellular arginase activity from neutrophils [21], but likely also from other mechanisms [38]. Future studies are needed to fully investigate a functional link between the depressed bacterial killing of monocytes recruited to CFASN and dysregulation of ISG15, CXCL11, NO, and interferon signaling in general, whose relationship to monocyte antibacterial activity is complex [15,28,39,40,41]. Interestingly, in a parallel effort to study transcriptional adaptations induced upon migration into CFASN in neutrophils, we found that broad changes caused by exposure to this milieu also caused a loss of killing ability toward *P. aeruginosa* in these cells [42]. Hence, the CF airway microenvironment negatively impacts incoming monocytes and neutrophils in their ability to kill bacteria, which may explain the paradoxical association of chronic airway inflammation with bacterial colonization in patients with CF.

While our study provides solid proof-of-concept data for the use of our model to transmigrate and condition primary monocytes in CF or healthy control airway fluid, it is limited in scope and will require follow-up studies to answer specific questions pertaining to CF monocyte biology. For example, since the origin of monocytes did not substantially impact functional responses (surface phenotype, metabolic activity, bacterial killing), transcriptional profiling of monocytes after transmigration to CFASN vs. HCASN was performed on HC blood monocytes. Nevertheless, we noticed variability in the responses measured for some outcomes (e.g., surface phenotype), which may reflect polymorphism in functional traits of human monocytes [43], highlighting the importance of conducting such studies with primary human cells, rather than animal models. Alternatively, some of this variability may represent a limitation of our transmigration model as applied to monocytes. Indeed, prior use of the model to transmigrate neutrophils did not yield such variability in functional responses [8]. Additionally, while the literature is split on whether CFTR function is critical or not to monocyte physiology [3,16,17,44], a follow-up study profiling CF blood monocytes upon transmigration to CFASN and HCASN would be interesting in this regard. For example, it is unclear whether CFTR may independently affect NO production, and other targets identified in our study (ISG15, CXCL11) are even less well characterized in CF pathophysiology in vivo and therefore warrant more attention. More broadly, the ability to mass-produce airway-like monocytes using this transmigration model should open opportunities for research into the roles played by these cells in chronic lung pathologies including but not limited to CF [45,46] and in acute infection by agents such as influenza or SARS-CoV-2 [47], which are known to modulate immune effectors such as NO [48] and ISG15 [49].

## 4. Materials and Methods

### 4.1. Study Population

Adult blood and sputum samples were collected following informed consent from 12 healthy controls and from 19 CF patients (Table 1) enrolled in the CF Biospecimen Repository (CFBR) within the Center for CF and Airways Disease Research at Emory University and Children’s Healthcare of Atlanta. The study was approved by our Institutional Review Board.

### 4.2. Blood Collection and Monocyte Isolation

Blood was collected by venipuncture in EDTA tubes, and monocytes were isolated by negative selection, using the RosetteSep isolation kit (Stem Cell Technologies, Vancouver, BC, Canada). Residual red blood cells were lysed via hypotonic shock, comprising a sequential 30-s incubation in ice-cold, filtered water, followed by an equal volume of 1.8% NaCl. Cells were centrifuged at 400× *g* for 10 min and monocyte yield and viability were determined using a hemocytometer following staining with ethidium bromide and acridine orange.

### 4.3. Transmigration Model

Purified monocytes were loaded in the in vitro transmigration model [8] as follows: (1) transmigration towards HCASN combined with the chemoattractant LTB4 (100 nM, Sigma, St Louis, MO, USA) mimicking recruitment to healthy, sterile airways; (2) transmigration towards CFASN (generated from CF sputum [8]), mimicking recruitment to CF airways. Monocytes were then collected at 10 h post-transmigration and used for downstream assays.

### 4.4. Flow Cytometry

Monocytes from blood (pre-transmigration and post-transmigration) were stained as described before [8], with the following modifications. Antibodies were directed against CD33 (Biolegend, San Diego, CA, USA, cat# 366614), CD36 (Biolegend, cat# 336204), CD47 (Biolegend, cat# 323114), CD63 (Biolegend, cat# 353026), CD66b (Biolegend, cat# 392904), CD91 (Invitrogen, cat# 46091942), CD172a (Biolegend, cat# 372106), HLA-DR (Biolegend, cat# 307626), PD-1 (Biolegend, cat# 329952), and PD-L1 (Biolegend, cat# 329738). Viability was determined based on staining with Live/Dead Violet (ThermoFisher Scientific, Wlatham, MA, USA, cat# L34958). Intracellular NO was measured with the cell-permeable probe diaminofluorescein-2-diacetate, as detailed before [5]. Samples were incubated with antibodies and viability or NO probe in the dark, at 4 °C for 30 min, washed with PBS-EDTA, and centrifuged at 400× *g* at 4 °C for 10 min, after which the supernatant was removed and samples were fixed overnight at 4 °C in Lyse/Fix buffer (BD Biosciences, San Jose, CA, USA, cat# 558049). Samples were resuspended in 300 µL PBS plus 10 µL counting beads (ThermoFisher Scientific, cat# C36950) to enable absolute cell counting during acquisition on a LSRII flow analyzer (BD Biosciences).

### 4.5. Bacterial Killing

Overnight cultures of *P. aeruginosa* and *S. aureus* were diluted and incubated in LB medium at 37 °C with shaking for 90 min, to obtain bacteria in the growth phase of the curve. After the concentration was determined at OD600, the bacteria were incubated at 37 °C in RPMI with 10% FBS for 30 min, while transmigrated monocytes were resuspended in RPMI with 10% FBS and incubated for 10 min at 37 °C, 5% CO_2_. Transmigrated monocytes and bacteria were co-incubated at MOI of 1 on a rotating wheel end-over-end at 37 °C, 5% CO_2_ for 1 h, after which cells were gently lysed with 0.1% Triton-X and serial dilutions were performed in sterile PBS. Bacteria were plated on agar plates and cultured for 12 h at 37 °C before colony counting.

### 4.6. Metabolic Analysis

Cells were freshly isolated from blood or harvested from transmigrated conditions after 10 h and resuspended in a pH-controlled Seahorse medium supplemented with glutamine (Agilent Technologies, Santa Clara, CA, USA). Next, cells were plated on XFp plates, at 2 × 10^5^ cells per well, and loaded onto an XFp analyzer. OCR was measured continuously as an index of mitochondrial respiration upon sequential injections of metabolic drugs, based on the provided Mito Stress Test kit, which includes oligomycin (ATP synthase/complex V inhibitor); fluoro-carbonyl cyanide phenylhydrazone (FCCP, ionophore that shunts the proton gradient across the mitochondrial membrane); rotenone and antimycin A (complex I and III inhibitors), to which we added an injection of 2-deoxyglucose (2-DG, hexokinase inhibitor shutting down glucose metabolism).

### 4.7. RNA Extraction and Transcriptomic Analysis

RNA was isolated utilizing the NucleoSpin RNA isolation kit (Takara Biosciences, Mountain View, CA, USA) and stored at −80 °C until use. RNA quality was assessed using the Bioanalyzer (Agilent Technologies), while RNA libraries were prepared following the Illumina True seq manufacturer’s protocol. Samples were subsequently run on the Nextseq 550 sequencing system at 25 million single-ended reads per sample. Produced Fastq files from single-end reads were aligned to the human reference genome (GRCh38.p13- Ensembl) using the alignment tool HISAT2 (version 2.1.0), using the default settings. Then, BAM files were sorted using SAMtools. Finally, to generate read counts expressed per gene, the tool FeatureCounts (1.5.2) was used. All processed counts were analyzed using DEseq2 to obtain differentially expressed genes (DEGs) between pre- and post-transmigration. DEGs were defined as genes with fold changes >2 folds, false discovery rate <0.1, and *p*-value <0.05. To understand the functions of the DEGs, the Metacore server was used against pathway maps and pathway networks. To identify unique dysregulations of gene expression, all DEGs were intersected between conditions using the UGent bioinformatic webtool for Venn diagrams (http://bioinformatics.psb.ugent.be/webtools/Venn/) (accessed on 23 February 2021). Subsequent enrichment of distinctive gene terms was conducted in the MSigDB v7.2 database (https://www.gsea-msigdb.org/gsea/index.jsp) (accessed on 23 February 2021) for molecular signature pathway outputs [50,51,52]. See online repository for transcriptomic comparisons presented in this manuscript at: https://data.mendeley.com/datasets/gcp66ch34c/3 (accessed on 23 February 2021).

## Figures and Tables

**Figure 1 ijms-22-02530-f001:**
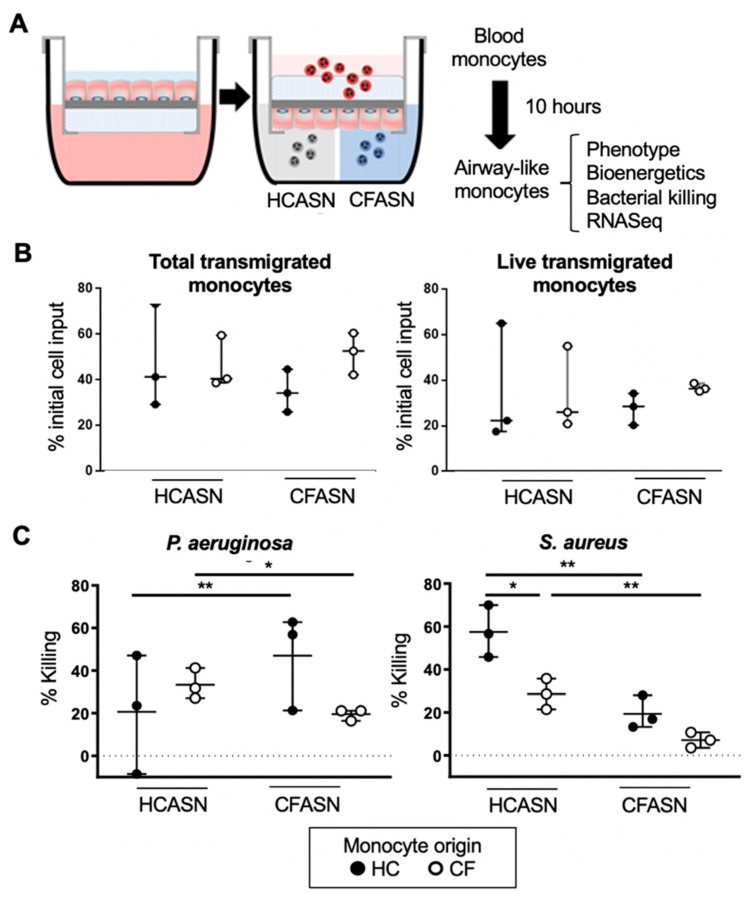
Yield and bactericidal activity of transmigrated monocytes in our model. (**A**) Our model enables transmigration of blood monocytes from healthy control (HC) or cystic fibrosis (CF) donors into HC or CF airway supernatant (HCASN or CFASN, mimicking recruitment into HC or CF airways, respectively), after which airway-like monocytes are analyzed functionally and transcriptionally. (**B**) Yields of total transmigrated monocytes (left) and live transmigrated monocytes (right) are shown as % of initial blood monocyte input. (**C**) Bactericidal activity of transmigrated monocytes against *P. aeruginosa* (left) and *S. aureus* (right) is shown as % of initial bacterial input. Statistics: Mann–Whitney test for different monocytes (HC vs. CF) recruited to the same ASN (HC or CF) and Wilcoxon signed rank test for the same monocytes (HC or CF) recruited to different ASN (HC vs. CF). * and ** mark *p* < 0.05 and *p* < 0.01, respectively. *n* = 3 per condition. Non-significant differences are not indicated.

**Figure 2 ijms-22-02530-f002:**
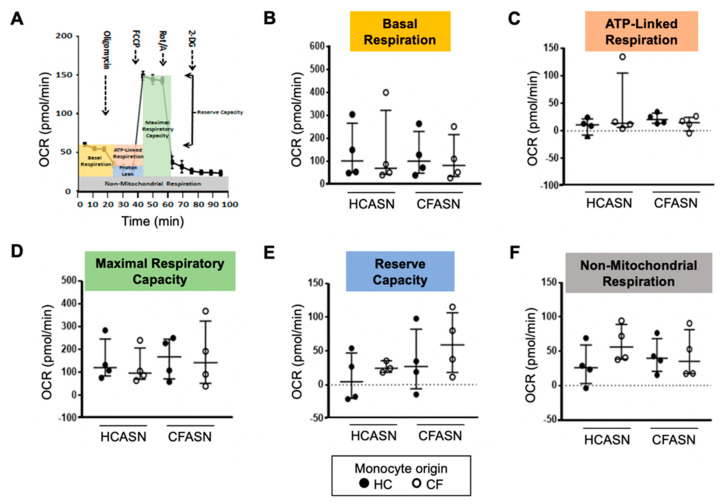
Metabolic activity of transmigrated monocytes. (**A**) Oxygen consumption rate (OCR) tracing upon sequential addition of oligomycin; fluoro-carbonyl cyanide phenylhydrazone (FCCP); rotenone and antimycin A (Rot/A); and 2-deoxyglucose (2-DG). OCR is used to compare (**B**) basal respiration, (**C**) ATP-linked respiration, (**D**) maximal respiratory capacity, (**E**) reserve capacity, and (**F**) non-mitochondrial respiration across types of transmigrated monocytes. Statistics: Mann-Whitney test for different monocytes (HC vs. CF) recruited to the same ASN; Wilcoxon signed rank test for the same monocytes recruited to different ASN (HC vs. CF). *n* = 4 per condition. Non-significant differences not shown.

**Figure 3 ijms-22-02530-f003:**
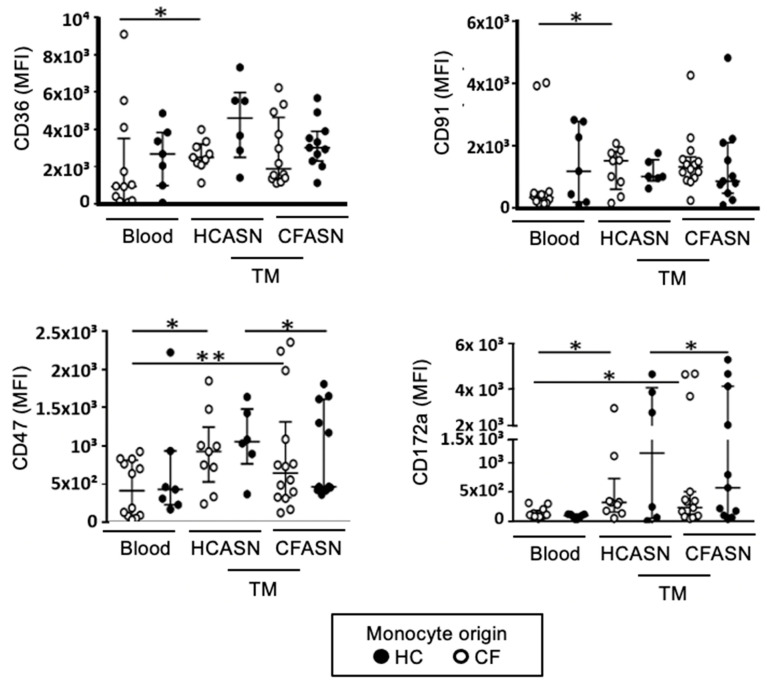
Expression of select scavenger receptors on transmigrated monocytes. Blood monocytes from HC or CF donors transmigrated into HCASN or CFASN were subjected to flow cytometric analysis to compare expression of scavenger receptors across subsets of transmigrated monocytes. Statistics: Mann-Whitney test for different monocytes (HC vs. CF) recruited to the same ASN (HC or CF) and Wilcoxon signed rank test for the same monocytes (HC or CF) recruited to different ASN (HC vs. CF). * and ** mark *p* < 0.05 and *p* < 0.01, respectively. *n* = 6–14 depending on condition. Non-significant differences not shown.

**Figure 4 ijms-22-02530-f004:**
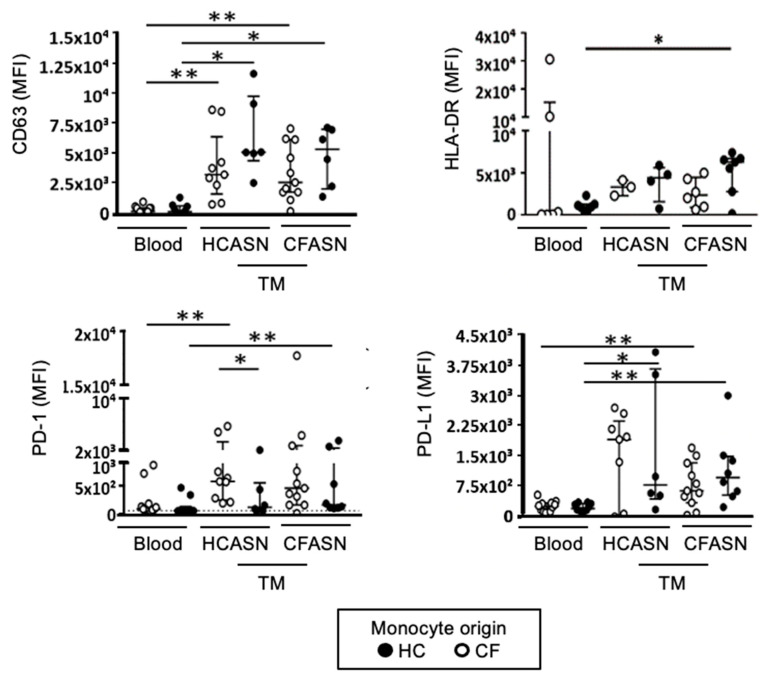
Expression of select activation markers on transmigrated monocytes. Blood monocytes from HC or CF donors transmigrated into HCASN or CFASN were subjected to flow cytometric analysis to compare expression of activation markers across subsets of transmigrated monocytes. Statistics: Mann–Whitney test for different monocytes (HC vs. CF) recruited to the same ASN (HC or CF) and Wilcoxon signed rank test for the same monocytes (HC or CF) recruited to different ASN (HC vs. CF). * and ** mark *p* < 0.05 and *p* < 0.01, respectively. *n* = 4–11 depending on condition. Non-significant differences not shown.

**Figure 5 ijms-22-02530-f005:**
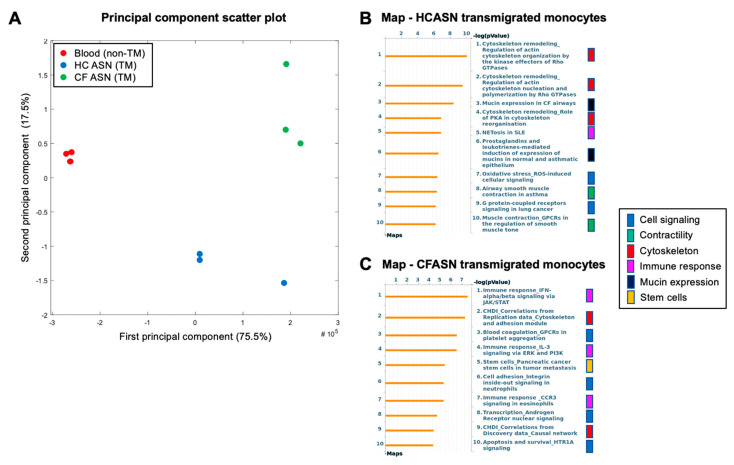
Transcriptomic analysis of transmigrated monocytes. (**A**) Principal component analysis of RNASeq data from healthy control blood monocytes prior to (non-transmigrated, red) and after transmigration (TM) to HC (blue) or CFASN (green) shows clear subsetting by condition (*n* = 3 independent biological replicates each). Pathway analysis by Metacore (right) illustrating differentially expressed genes and pathways in monocytes transmigrated to (**B**) HCASN and (**C**) CFASN, respectively, compared to blood monocytes.

**Figure 6 ijms-22-02530-f006:**
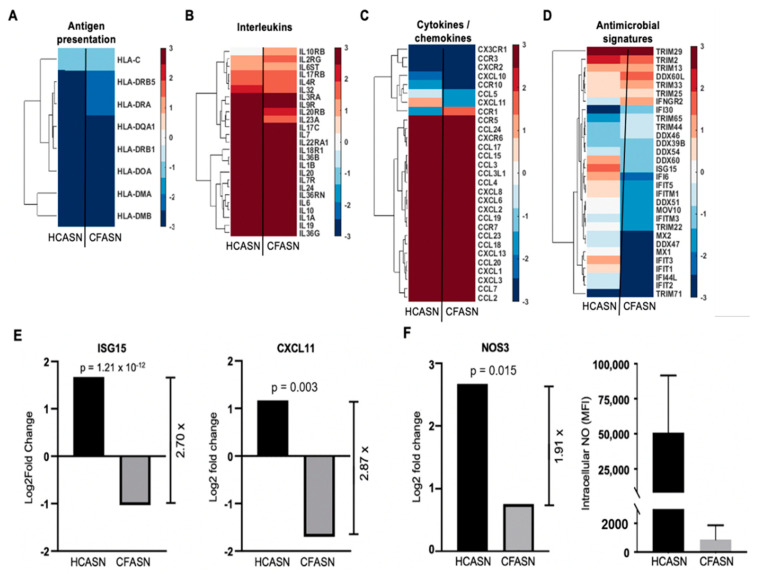
Analysis of differentially expressed genes (DEGs) related to immunity in monocytes transmigrated to HCASN vs. CFASN. Shown are chosen DEGs representative of (**A**) antigen presentation, (**B**) interleukins, (**C**) cytokines/chemokines, and (**D**) antimicrobial signatures. Shown in (**E**) are ISG15 and CXCL11, two genes with direct antibacterial functions, which are significantly downregulated in CFASN-recruited monocytes (Log2 scale for both), and in (**F**) the nitric oxide producing enzyme NOS3 (left panel, Log2 scale) and corresponding intracellular NO levels (right panel, shown as MFI).

**Table 1 ijms-22-02530-t001:** Subject demographics. F, female; HO, patient with CF homozygous for F508Del mutations; HZ, patient with CF carrying one F508Del and one other mutation; M, male; OT: patient with CF carrying two mutations other than F508Del.

Participant Group	N	CFTR Genotype	Gender	Age (Years)
Healthy Control	12	WT	F = 7, M = 5	25.9 ± 1.5
CF Patient	19	HO = 9, HZ = 7, OT = 3	F = 8, M = 11	31.2 ± 2.9

## Data Availability

Data supporting reported results can be found in an online repository at: https://data.mendeley.com/datasets/gcp66ch34c/3 (accessed on 23 February 2021).

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
