# Peer review of "Functional and Transcriptional Adaptations of Blood Monocytes Recruited to the Cystic Fibrosis Airway Microenvironment In Vitro"

_ijms, 2021, doi:10.3390/ijms22052530_

Round 1

Reviewer 1 Report

In this manuscript, Ford and colleagues investigate possible monocytes adaptations occurring upon their recruitment to CFASN as compared to HCASN. In previous studies, the authors demonstrated that blood neutrophils (from healthy subjects) migrated through well-differentiated lung epithelium into CFASN mimic the dysfunction of CF airway neutrophils in vivo, including decreased bactericidal activity despite an increased metabolism. Thus, the question whether monocytes undergo a similar fate is relevant.

The author evaluated surface phenotype, metabolic and bacterial killing activities, and transcriptomic profile of monocytes pre- and post-transmigration. They found that monocytes were not metabolically activated nor showed broad differences upon recruitment to CFASN compared to HCASN. However, monocytes recruited to CFASN showed decreased bactericidal activity, which could explain why they also fail to protect CF airways from infection.

RNASeq analysis showed  differenced on monocyte RNA profile following CFASN vs HCASN conditions, particularly in immune and antiviral signaling.

This work pave the ways to further studies to establish whether CF blood monocytes behave similarly or whether they show more evident changes for example in surface phenotype and metabolic features. Indeed, the relevance of CFTR dysfunction in monocytes biology is still debated, thus the existence of differences in their adaptations to ASN having different characteristics is not sure. 

Author Response

R1- In this manuscript, Ford and colleagues investigate possible monocyte adaptations occurring upon their recruitment to CFASN as compared to HCASN. In previous studies, the authors demonstrated that blood neutrophils (from healthy subjects) migrated through well-differentiated lung epithelium into CFASN mimic the dysfunction of CF airway neutrophils in vivo, including decreased bactericidal activity despite an increased metabolism. Thus, the question whether monocytes undergo a similar fate is relevant.

The author evaluated surface phenotype, metabolic and bacterial killing activities, and transcriptomic profile of monocytes pre- and post-transmigration. They found that monocytes were not metabolically activated nor showed broad differences upon recruitment to CFASN compared to HCASN. However, monocytes recruited to CFASN showed decreased bactericidal activity, which could explain why they also fail to protect CF airways from infection.

RNASeq analysis showed differences in monocyte RNA profile following CFASN vs HCASN conditions, particularly in immune and antiviral signaling.

This work paves the way to further studies to establish whether CF blood monocytes behave similarly or whether they show more evident changes for example in surface phenotype and metabolic features. Indeed, the relevance of CFTR dysfunction in monocytes biology is still debated, thus the existence of differences in their adaptations to ASN having different characteristics is not sure.

C1- We thank the reviewer for this positive appraisal. We added a sentence in the Discussion to further comment on the question of the relevance of CFTR dysfunction in monocytes.

Reviewer 2 Report

The manuscript by Ford et al., al test the hypothesis that blood monocytes undergo adaptations when exposed to Cystic Fibrosis (CF) airway fluid supernatant (CFASN), based on their prior observations of blood neutrophils under similar conditions. To support this hypothesis, the authors used an in vitro transmigration model, using human blood monocytes obtained from healthy control or CF subjects, to condition them in airway fluid. They found that although the monocytes transmigrated to the CFASN do not differ in surface phenotype and metabolism, they do show reduced bacterial killing capacities. The authors further did a transcriptome analysis to show some subtle differences in antiviral genes. Overall, the manuscript was started on a brighter note, but failed to justify the rationale. Reduced bacterial killing is the only finding in this manuscript and subtle changes in antiviral gene signatures cannot explain the outcome.

Major comments:

  1. Flow plots and histograms must be shown for each panel based on which MFI was calculated. Also, I am surprised to see the huge variation (3-4-fold) in the same group (for ex., metabolic activity or scavenger receptors) which may have resulted from some technical challenges of the model system and thus lessen the significance, if any.
  2. Figures are of very poor quality and lot of aspects of gene expression are not visible.
  3. Supplementary Fig. 1 is not at all helpful. The way it can be shown is, as an excel table with the comparison of the rpkm/fpkm values and the fold difference. GO can be included in the same.
  4. The discussion seems like an extension of results. The authors should discuss clearly the gap in the field and how their study will help to move the field forward.
  5. 2C and 2D are mislabeled in the text. No “n” numbers were provided for the in vitro data.

Author Response

R2- The manuscript by Ford et al., al tests the hypothesis that blood monocytes undergo adaptations when exposed to Cystic Fibrosis (CF) airway fluid supernatant (CFASN), based on their prior observations of blood neutrophils under similar conditions. To support this hypothesis, the authors used an in vitro transmigration model, using human blood monocytes obtained from healthy control or CF subjects, to condition them in airway fluid. They found that although the monocytes transmigrated to the CFASN do not differ in surface phenotype and metabolism, they do show reduced bacterial killing capacities. The authors further did a transcriptome analysis to show some subtle differences in antiviral genes. Overall, the manuscript was started on a bright note, but failed to justify the rationale. Reduced bacterial killing is the only finding in this manuscript and subtle changes in antiviral gene signatures cannot explain the outcome.

C2- We thank the reviewer for this insightful critique. We agree that the initial manuscript could be interpreted as providing only a faint connection between the finding of a defective bacterial killing function in monocytes recruited to CFASN and the changes observed in their transcriptomic profile. To clarify this issue, we corrected the labeling of genes found to be downregulated in CFASN-recruited monocytes from “antiviral” to “antimicrobial”. Indeed, this set included genes with strong impact on immune cell antibacterial responses like ISG15 and CXCL11 which were significantly downregulated in CFASN-recruited compared to HCASN-recruited. In a prior study of P. aeruginosa infection (PMID: 32416603, now cited in the Discussion), ISG15 downregulation alone was sufficient to explain decreased bacterial killing. Thus, the genes identified as being differentially regulated by CFASN exposure may explain the defective killing of bacteria in this condition. We further emphasized this point by including bar plots for ISG15, CXCL11 and other genes significantly changes in CFASN vs. HCASN conditions, along with fold changes and p values (new Fig. 5B).

R3- Flow plots and histograms must be shown for each panel based on which MFI was calculated.

C3- Flow plots representing the gating strategy for monocytes were included in the initial supplement and have been kept (Fig. S1). To fully address the reviewer’s request, we included in the revised manuscript representative histograms for each of the 8 markers measured on monocytes (Figs. 3 and 4). These are now shown in two corresponding supplementary figures (new Figs. S2A and S2B).

R4- Also, I am surprised to see the huge variation (3-4-fold) in the same group (for ex., metabolic activity or scavenger receptors) which may have resulted from some technical challenges of the model system and thus lessen the significance, if any.

C4- We thank the reviewer for bringing up this point. We added a comment on data variability in the Discussion. While we acknowledge the significant magnitude of some of the variations in response observed in our model for some of the measured outcomes, this may be accounted for at least in part by inherent polymorphisms in human monocytes. Indeed, monocytes have been under strong selective pressure in humans, leading to considerable variation in responses to stimuli, as exemplified by PMID: 25327457 (now cited in the Discussion). This highlights the importance of conducting investigations with primary human cells, as featured in our model.

R5- Figures are of very poor quality and lot of aspects of gene expression are not visible.

C5- We take this critique to reflect a lack of readability of the two volcano plots illustrating gene expression in CFASN- and HCASN-recruited monocytes in Fig. 5 of the initial manuscript. To improve quality of these volcano plots, we now included them as full size figures in the revised manuscript supplement (Figs. S3 and S4), which makes individual genes featured in the plots easily readable. The new Fig. 4 now features a PCA plot (now Fig. 4A, formerly Fig. S2) and the maps of differentially expressed genes (now Fig. 4B and C, formerly bottom parts of Fig. 4A and B).

R6- Supplementary Fig. 1 is not at all helpful. The way it can be shown is, as an excel table with the comparison of the rpkm/fpkm values and the fold difference. GO can be included in the same.

C6- We take it that the reviewer is referring to Table S1 (which listed families differentially expressed genes) rather than Fig S1 (which showed flow gating). To comply with this critique, we eliminated Suppl Table1 and instead generated Excel tables with fold differences in the 3 following comparisons: HCANS vs. blood, CFASN vs. blood, and HCASN vs. CFASN. These tables are in a Mendeley data repository, providing full explanations for readers interested in further mining our datasets (posted at: https://data.mendeley.com/datasets/gcp66ch34c/2).

R7- The discussion seems like an extension of results. The authors should discuss clearly the gap in the field and how their study will help to move the field forward.

C7- We revised the Discussion to address this important critique. We attempted to discuss the gaps in

R8- 2C and 2D are mislabeled in the text. No “n” numbers were provided for the in vitro data.

C8- We thank the reviewer for noticing this and made corrections as needed.

Round 2

Reviewer 2 Report

The manuscript has been substantially improved. Figure quality still needs to be improved (Figure 1-4 looks really sketchy and pixelated).